# The Evaluation of Inflammatory Biomarkers in Predicting Progression of Acute Pancreatitis to Pancreatic Necrosis: A Diagnostic Test Accuracy Review

**DOI:** 10.3390/healthcare11010027

**Published:** 2022-12-22

**Authors:** Hafiz Muhammad Asim Riaz, Zara Islam, Lubna Rasheed, Zouina Sarfraz, Azza Sarfraz, Karla Robles-Velasco, Muzna Sarfraz, Ivan Cherrez-Ojeda

**Affiliations:** 1Department of Research, Services Institute of Medical Sciences, Lahore 54000, Pakistan; 2Department of Research, Sahiwal Medical College, Sahiwal 57040, Pakistan; 3Department of Research, University of Medical and Dental College, Faisalabad 38800, Pakistan; 4Department of Research and Publications, Fatima Jinnah Medical University, Lahore 54000, Pakistan; 5Department of Pediatrics and Child Health, The Aga Khan University, Karachi 74800, Pakistan; 6Department of Allergy and Pulmonology, Universidad Espíritu Santo, Samborondón 092301, Ecuador; 7Department of Research, King Edward Medical University, Lahore 54000, Pakistan

**Keywords:** inflammatory biomarkers, acute pancreatitis, necrosis, c-reactive protein, calcitonin, lactate dehydrogenase, diagnostic test accuracy, healthcare, medicine

## Abstract

Acute pancreatitis is the acute inflammation of the pancreas; 30% of cases may progress to pancreatic necrosis. The aim of this study was to assess the diagnostic accuracy of inflammatory biomarkers (C-reactive protein (CRP), procalcitonin (PCT), and lactate dehydrogenase (LDH)) in detecting pancreatic necrosis in adults with confirmed acute pancreatitis within 14 days of symptom onset and without organ failure. A systematic search was conducted across the Cumulative Index of Nursing and Allied Health (CINAHL), Cochrane, Embase, PubMed, and Web of Science until May 30, 2022, with the following keywords: acute disease, biomarkers, C-reactive protein, calcitonin, differential, diagnosis, lactate dehydrogenase, pancreatitis, acute necrotizing, necrosis, sensitivity, specificity. Statistical analysis was conducted in RevMan 5.4.1 (Cochrane). Five studies pooling 645 participants were included of which 59.8% were males, with a mean age of 49 years. CRP was the best cutoff at 279 mg/L (χ2 = 47.43, *p* < 0.001), followed by 200 mg/L (χ2 = 36.54, *p* < 0.001). LDH was cut off at 290 units/L (χ2 = 51.6, *p* < 0.001), whereas PCT did not display the most reliable results at 0.05 ng/mL. Inflammatory biomarkers are scalable diagnostic tools that may confer clinical value by decreasing the mortality of acute pancreatitis sequelae.

## 1. Background

Acute pancreatitis (AP) is an acute disorder of the exocrine pancreas, typically associated with acinar cell injury and local/systemic inflammatory responses [1,2,3]. AP ought to be suspected among patients that present with severe acute pain in the mid-epigastrium or the left upper quadrant radiating to the back [4,5]. However, biochemical (serum amylase or lipase 3x ULN) or radiological evidence, such as computed tomography (CT), is required for diagnosis [6]. A history of alcohol intake, cholelithiasis, metabolic disorders, malignancy, or infection is associated with the incidence of AP [6]. The severity may range from a mild disease involving minimal edema in the pancreas, which recovers fully in a few days, to severe disease that is typified by persistent inflammatory responses with or without multiple organ failure—with or without peripancreatic or pancreatic necrosis (PN) [6,7,8]. With over 200,000 hospital admissions owing to AP in the United States, a multidisciplinary approach to diagnosis and management is compulsory [9,10,11]. A common pancreatic complication of AP is the collection of pancreatic fluid, which is defined by the presence or absence of necrosis defined by the 2012 Revised Atlanta classification [12,13,14,15,16]. In the case that necrosis occurs in the form of necrotizing pancreatitis (~30% of patients with AP), acute necrotic collections are formed within the first four weeks (<4 weeks) [17,18], whereas walled-off necrosis comprising encapsulated collections may be formed after four weeks have lapsed (>4 weeks) [19,20]. Infected necrotizing pancreatitis carries a worse prognosis as compared to sterile necrotizing pancreatitis, with in-hospital mortality of 30% for infected groups [15,16,21,22]. 

Among individuals with established AP, inflammatory biomarkers, namely C-reactive protein (CRP), serum procalcitonin (PCT), and serum lactate dehydrogenase (LDH), may help in the early diagnosis of pancreatic necrosis (PN) because radiological findings can take several days or weeks to establish [23,24,25,26]. In the event that a diagnosis of AP is made, biochemical testing is routinely performed; with serum amylase being utilized as a common biochemical marker for the diagnosis, its sensitivity is reduced by late presentation, chronic alcoholism, and hypertriglyceridemia [27]. The caveat is that CRP is an inflammatory biomarker that may be elevated due to ongoing tissue damage in PN; however, CRP may also rise with edematous pancreatitis [28]. Radiological examination (i.e., contrast-enhanced, CT) is required to distinguish edematous pancreatitis and necrotizing pancreatitis in such cases [29,30]. At present, CRP has been considered as a triage test to identify AP among patients without organ failure; these patient groups are then radiologically tested if the CRP test values indicate necrosis [31]. PCT is a protein typically found in the thyroid, but all tissues can produce it; patients with severe inflammation and sepsis have elevated PCT levels [32,33]. Since PN is directly correlated to ongoing inflammation, PCT may serve as a utilizable biomarker in distinguishing necrotizing pancreatitis and edematous pancreatitis [34]. PCT is typically undetectable in healthy adults; hence, the presence of PCT coupled with an increasing trend of other laboratory values may be scalable in triaging patients who require radiological examination for PN [35]. LDH is an indicator of cell death, and since PN is associated with cell death, it may be also of clinical value in identification [36]. 

The curative pathway of patients with AP varies between those with or without PN and underlying organ failure [37]. AP patients with organ failure are more likely to be radiologically evaluated [38]. In case necrotizing pancreatitis is suspected among patients without organ failure, laboratory testing of the inflammatory biomarkers may be conducted. However, the thresholds/cutoffs of the inflammatory biomarkers in formulating diagnosis and monitoring progression to PN once AP is diagnosed remain elusive. The role of CRP, LDH, and PCT, on the whole, requires deliberation to improve hematological and gastroenterological society guidelines worldwide. This diagnostic test accuracy review has been conducted to gain a better understanding of the three inflammatory biomarkers and the cutoffs required to either proceed with radiological investigation or intensive care therapy. We aimed to assess the diagnostic accuracy of inflammatory biomarkers (CRP, procalcitonin, and LDH) in detecting PN in adult patients with AP within 14 days of symptom onset (acute phase) and without organ failure. 

## 2. Methods

This diagnostic test accuracy review was conducted in accordance with the Preferred Reporting Items for Systematic Reviews and Meta-Analysis (PRISMA) 2020 Checklist [39]. In accordance with the review question, this paper sought to assess the sensitivity and specificity of index tests (CRP, PCT, LDH) in predicting the progression of acute pancreatitis to pancreatic necrosis. 

### 2.1. Searches

Study searches were conducted across the following databases: PubMed (MEDLINE), Cumulative Index of Nursing and Allied Health (CINAHL) Plus (EBSCO host), EMBASE (Elsevier), Cochrane Central (Wiley Online Library), and Web of Science (Clarivate Analytics) until 30 May 2022. An umbrella methodology was applied where a manual search of reference lists of all included studies was conducted. The search strategy was developed with the expertise of three reviewers who are mid-to-late-career researchers (Z.S., A.S., and I.C.-O.). The search terms were adjusted to the specificities of the different databases. Either database-specific subject headings (i.e., MeSH) or keyword-specific applications were made with the Boolean logic (and/or). Keywords were utilized as follows: “Acute Disease, Biomarkers, C-Reactive Protein, Calcitonin, Differential, Diagnosis, Lactate Dehydrogenase, Pancreatitis, Acute Necrotizing, Necrosis, Sensitivity, Specificity.” The search strategy for PubMed is appended in Table 1. No time restrictions were applied to ensure all relevant studies were located. Additionally, all non-English studies were translated into English using open-source software (i.e., Google Translate).

### 2.2. Inclusion and Exclusion Criteria

Studies were included if they evaluated the accuracy of index tests. Only clinical trials and cohorts were included to minimize the high risk of biases. Case reports and case series were excluded owing to innate risks in study design, population, and sampling. The participants were required to have acute pancreatitis within two weeks of the onset of symptoms; the interval between the onset of symptoms and the timing of the test was irrelevant, and the onset of symptoms was documented to be 14 days, which is the acute phase of infection. The cutoff is typical in diagnostic test accuracy reviews, as recommended by the Cochrane Handbook [40]. A diagnosis of acute pancreatitis was required to be made based on existing guidelines, at the time of paper curation, as enlisted in the studies. In the case that the participants developed organ failure/required intensive care, they were excluded since they had already undergone radiological tests such as CT and MRI. Additionally, participants that already presented with pancreatic necrosis on the CT scan were excluded. Many studies were excluded due to the lack of interpretable test results for diagnostic test accuracy. The full breakdown is given under the PRISMA flowchart, depicting the study selection process. 

#### Case Definitions

This diagnostic test accuracy review predicts the progression of acute pancreatitis to pancreatic necrosis, defined as follows:***Acute Pancreatitis*** is the sudden inflammation of the pancreas that may be mild or life-threatening but usually subsides [41].***Necrotizing pancreatitis*** occurs when pancreatic tissue dies due to inflammation [42].

### 2.3. Inflammatory Biomarkers (Index Tests) and Disease Outcomes

CRP, Procalcitonin, and LDH in combination or alone, before the conduction of radiological tests will be annotated as ‘index tests’ or referred to as inflammatory biomarkers in this text. Many kits are available to measure the level of these inflammatory biomarkers, from a variety of manufacturers worldwide. The contents of this study will include studies regardless of the thresholds used across the studies, and they will be listed if they deviate from known standards. The studies either performed single or sequential tests of these inflammatory biomarkers. The target disease is pancreatic necrosis, which may be either peripancreatic, sterile pancreatic, or infected necrosis. Certain lags were seen in the inflammatory biomarker testing period and the radiological investigations that were also performed within 24 h of the organ failure diagnosis. However, even then, pancreatic necrosis does not resolve within 24 h, and the laboratory values and cutoffs were not impacted.

### 2.4. Data Extraction and Synthesis

Two authors (Z.S. and A.S.) extracted data together into a shared spreadsheet with the final author (I.C.-O.) present for any disagreements. The data were extracted as “author/year, study design, participant count, male/female, inclusion/exclusion criteria, mean age, the time between the onset of symptoms and index test, etiology of acute pancreatitis, percentage of participants with necrosis, index tests performed, thresholds used in index tests, reference standard.” The true positives (TP), false positives (FP), false negatives (FN), and true negatives (TN) were also listed. In the case that one study reported multiple cutoffs of the same inflammatory biomarker test, TP/FP/FN/TN values were computed at any of the different thresholds. The data analysis was conducted in Review Manager 5.4.1 (RevMan, Cochrane), with additional computations conducted for the following [43]:**Positive Predictive Value (PPV)** = (True Positives)/(True Positives + False Positives);**Negative Predictive Value (NPV)** = (True Negatives)/(True Negatives + False Negatives);**Positive Likelihood Ratio** = Sensitivity/(1 − Specificity);**Negative Likelihood Ratio** = (1 − Sensitivity)/Specificity.

### 2.5. Risk of Bias

Two authors (Z.S. and A.S.) conducted the quality assessment of included studies utilizing the QUADAS-2 assessment tool– this tool is solely created for diagnostic test accuracy reviews [44]. Studies were considered to be of high methodological quality if they had a low concern and low risk of bias. The findings were presented under risk of bias summary and graphs with figure legends.

## 3. Results

### 3.1. Search Process

The PRISMA flow diagram is depicted in Figure 1. During the identification phase, 4926 studies were identified from all the databases. Of these, 1568 were duplicates. During the screening phase, 3358 studies were screened, and 2926 studies were excluded as the titles and abstracts did not meet the objectives of this study; during the full-text screening phase, 432 were assessed for eligibility. Of these, 276 did not provide diagnostic test data on pancreatic necrosis, 139 were non-original studies, and 12 were case reports. Finally, in the inclusion phase, five studies were added to this synthesis (Figure 1).

### 3.2. Overview of Included Studies

In total, five primary clinical studies cited across seven references were included in this analysis: Bertsch (1997) [45], Rau (1998) [46,47,48], Alfonso (2003) [49], Hagjer (2018) [50], and Vasudevan (2018) [51]. The five primary clinical studies pooled 645 participants who met the inclusion criteria and provided data on the diagnostic accuracy of the inflammatory tests among participants that had pre-established acute pancreatitis. The overall average age of participants was 48.97 years, where 386 (59.8%) were males. All the included studies were cohorts (3 were prospective, 1 retrospective, and 1 not specified). None of the studies reported a previous history of pancreatic disease, owing to no medical records in the care center or the patients’ unconscious state. Overall, the presentations comprised acute pancreatitis in the secondary or tertiary care setting as enlisted in Table 2. The various inflammatory biomarkers along with their sensitivities and specificities are also enlisted in the table. For PCT, both Bertsch (1997) and Hagjer (2018) employed a cutoff of 0.5 ng/mL, whereas, for CRP, cutoffs comprised 140 mg/L by Rau (1998), 200 mg/L, and 279 mg/L by Alfonso (2003), with 150 mg/L by Hagjer (2018) and 98 mg/L by Vasudevan (2018). Two studies were conducted in Germany, with two others in India, and one in Spain. The reference standards comprised CT scans in all studies with additional interoperative findings by Rau (1998) (Table 2).

Vasudevan (2017) reported on the etiologies (alcoholic, post-ERCP, biliary) of acute pancreatitis; Hagjer (2018) reported etiologies as being equally distributed between gallstones and alcohol. Bertsch (1997) and Alfonso (2003) stated that acute pancreatitis was of varied etiology, whereas Rau (1998) did not specify the etiology. Considering the presence of the target condition/disease, Rau (1998) indicated the presence of peripancreatic and pancreatic necrosis; Vasudevan (2017) and Hagjer (2018) indicated the condition as infected pancreatic necrosis, whereas the remaining studies did not provide further information on the necrosis type. Alfonso (2003) did not mention the exact day that CRP was measured, with two cutoffs including 200 mg/L and 279 mg/L. CRP was obtained on the day of admission by Vasudevan et al. at a cutoff value of 98 mg/L; within 1 day of presentation of symptoms by Hagjer et al. at a cutoff value of 150 mg/L; and on day 3 by Rau et al. at a cutoff value of 140 mg/L. Bertsch et al. tested PCT on day 1 with a cutoff value of 0.5 ng/mL; Hagjer et al. tested PCT within 1 day of symptom onset at a cutoff value of 0.5 ng/mL. LDH was assessed by Rau et al. on day 5 of symptom onset with a cutoff value of 290 U/L (Table 2).

### 3.3. Analytical Findings

The studies reported findings based on different thresholds. The sensitivities and specificities were computed and reported with 95% confidence intervals, reported as forest plots. Additional findings including PPV, NPV, positive likelihood ratio, and negative likelihood ratio are appended in Table 3.

Alfonso and colleagues reported CRP findings on an unknown day at a threshold of 200 mg/L, including 157 patients. The sensitivity was 0.88 (95% CI = 0.69–0.97) and the specificity was 0.75 (95% CI = 0.67–0.82). The study’s second cutoff of 279 mg/L for CRP had a sensitivity of 0.72 (95% CI = 0.51–0.88). Hagjer and colleagues, using a cutoff of 150 mg/L, had a sensitivity of 0.29 (95% CI = 0.11–0.52) and a specificity of 0.82 (95% CI = 0.66–0.92) when performing the test within a day. Rau and colleagues employed a cutoff of 140 mg/L on day 3 had a sensitivity of 0.82 (95% CI = 0.66–0.92) and a specificity of 0.84 (95% CI = 0.66–0.95). Vasudevan and colleagues employed a cutoff of 98 ng/mL on the day of admission and yielded a sensitivity of 0.67 (95% CI = 0.56–0.76) and a specificity of 0.67 (95% CI = 0.61–0.73) (Figure 2).

Bertsch et al. and Hagjer et al. both conducted PCT tests within one day of admission, however, presenting differing findings. Bertsch yielded a sensitivity of 0.75 (95% CI = 0.35–0.97) and a specificity of 0.57 (95% CI = 0.18–0.9), whereas Hagjer yielded a sensitivity of 0.28 (95% CI = 0.15–0.45) and a specificity of 0.81 (95% CI = 0.58–0.95). Rau and colleagues assessed LDH at a threshold of 290 U/L on day 5 and yielded a sensitivity of 0.87 (95% CI = 0.73–0.96) and a specificity of 1 (95% CI = 0.89–1) (Figure 3).

The summary receiver operating characteristic (SROC) plot showcases that LDH (day 5; >290 U/L) provides an excellent prediction of pancreatic necrosis. Furthermore, CRP at a cutoff of 140 mg/L provides excellent estimates with higher specificity than sensitivity, with a CRP level of 200 mg/L being more sensitive than specific. PCT findings belonged to a good range of prediction; however, they did not yield positive results when used as inflammatory biomarkers (Figure 4).

### 3.4. Methodological Assessment of the Included Studies

The methodological quality of included studies is summarized in Figure 5 and Figure 6.

Concerning the participant selection, 75% of the studies were at unclear risk of bias and 40% of them were of unclear concern about applicability in this domain. However, 60% had a low risk of bias when considering applicability, meaning that the participants were appropriately excluded and that random selection was employed in more than half of the included studies.

When assessing the index test, two studies had a high risk of bias, two had an unclear risk, and one had a low risk of bias. The thresholds were defined by three studies at a prespecified time; however, two studies did not clarify the specification of the index test. However, when considering the applicability of index tests, only one study had a high risk of bias whereas two each had a low risk of bias or unclear risks. All the studies reported the threshold at which the diagnosis was made, hence the applicability must be considered reliable.

The reference standard assessment revealed that four studies had an unclear risk of bias, whereas one had a low risk of bias. All the studies had a reference standard including a CT scan alone or a combination of laparotomy findings. When assessing all the studies, there were very low concerns about the applicability of the reference standard since the target condition across all studies was peripancreatic or pancreatic necrosis, both being the pre-defined target conditions.

For flow and timing, four studies had a low risk of bias, whereas one had a high risk of bias. The study with a high risk of bias did not define the exact day of conducting the test, whereas the other studies specified the exact day of conducting the test.

## 4. Discussion

Five studies comprising 645 participants were included in our diagnostic test accuracy review, where the diagnostic accuracy of inflammatory biomarkers was tested among patients diagnosed with AP. All five studies reported diagnostic test accuracies at different time points and with different thresholds. It is useful to view this study as an adjuvant to current clinical practice as CT scans expose patients to unrequired radiation exposure [52], particularly if patients have undergone radiological testing for AP recently. Studies with significant findings reported CRP cutoffs at 140 mg/L (*p* < 0.001) [46,47,48], 200 mg/L and 279 mg/L (*p* < 0.001) [49], and 98 ng/mL (*p* < 0.001) [51]. In addition, LDH had excellent positive predictive values (1, 95% CI= 0.903–1) at a cutoff of 290 U/L (*p* < 0.001) [46,47,48]. These triage tests are useful, particularly when they have high sensitivity and a reasonable bound of specificity. In our analysis, the sensitivities of the tests were various, but LDH upwards of 0.5 ng/mL and CRP above 200 mg/L had high sensitivities within 1–5 days of symptom onset.

The 2019 World Congress of Emergency Surgery (WCES) guidelines deliberate on the outcomes of peripancreatic and pancreatic necrosis [5]. The guidelines report that the mortality rate of infected necrosis with organ failure was reportedly 35.2% in a sample of 6970 patients, whereas concomitant organ failure and sterile necrosis had a mortality rate of 19.8% [5,53]. If the patients had infected necrosis without any signs of organ failure, the mortality was reported to be 1.4%. In a majority of patients with AP, CT scans may not be required, as the extension of PN can be noted with contrast-enhanced CT after 3 days of onset. However, the WCES guidelines state concerns over post-contrast acute kidney injury [54]. The WCES guidelines observe PCT as the most sensitive predictor for pancreatic infection with strong NPV for infected necrosis. Other studies state that PCT and LDH are also useful in predicting PN. PCT values of 3.8 ng/mL or higher within 96 h of the onset of symptoms have conferred a sensitivity of 93% and specificity of 79% [8,55]. Supporting information such as hypotension and the APACHE II scores at 1 day of admission may be independent predictors of infected necrosis as stated by Thandassery and colleagues based on findings employing 81 patients [56]. Values of blood urea nitrogen, hematocrit, lactate, and creatinine, which are markers of hypovolemia and tissue perfusion, ought to also be monitored to ensure the best outcomes [5].

### 4.1. Limitations and Strengths

Our study has a few limitations. In our included studies, reference standards were based on either histological confirmation of radiological deliberation of pancreatic necrosis (CT or contrast-enhanced MRI). This may have omitted some cases of pancreatic necrosis in our study. When ranking reference tests, a biopsy is considered the gold standard; however, it is less likely to be performed with negative laboratory tests for pancreatic necrosis. In addition, the exclusion of patients based on a specific threshold, particularly those at borderline, may overestimate the diagnostic accuracy of the tests. With Rau et al. and Hagjer et al.’s studies yielding similar PCT thresholds upon 70 and 60 patients, respectively, the sensitivity was significantly lower in the study by Rau et al.; there are limitations to the reliability of measuring or timing.

Our study also has certain strengths. The included studies did not restrict patients based on the etiologies of acute pancreatitis; hence, this study applies to all etiologies of AP. We conducted a thorough database search and included non-English studies without any date restrictions. All screened studies were scanned in full to not exclude relevant studies. The inclusion of these studies decreased the impact of publication bias if any in this diagnostic test accuracy review. The omitting of the case series was planned to reduce the risks of bias. All references were screened independently to limit inter-reviewer errors and the chance element. We analyzed an imperative disease with a global burden; the methodological quality of this study can be utilized by future researchers who wish to interpret the evidence of pancreatic inflammation and subsequent necrosis.

### 4.2. The Clinical Relevance of This Diagnostic Test Accuracy Review

Severe acute pancreatitis leads to pancreatic necrosis in 15–25% of patients with acute pancreatitis. As such, it represents one of the most severe complications of severe acute pancreatitis with a mortality rate of 20–30% [57].

We aimed to posit inflammatory biomarker testing in the laboratory, radiological testing, and surgical modalities as a viable means of standardizing diagnosis. Considerable advances have been made in imaging modalities for the diagnosis and management of acute pancreatitis. However, inappropriate radiological testing can increase costs on health systems, expose patients to radiation, and increase complication rates without conferring benefits to patients.

Contrast-enhanced CT is considered the diagnostic standard for the evaluation of AP due to the success the test has had in predicting severity and diagnosis. Nonetheless, it can be complemented by robust non-radiological, laboratory-based triage tests. Future studies are recommended to form a stronger causality linkage between AP and PN with the incorporation of laboratory-based tests. Both clinical judgment and tools to assess the severity and prognosis of disease are required to reduce morbidity and mortality due to post-pancreatic necrotic inflammation.

## 5. Conclusions

We assessed three tests, namely CRP, PCT, and LDH, all within two weeks of symptom onset of acute pancreatitis. The cutoff with optimal CRP value was 200 mg/L which had higher sensitivity, as compared to a cutoff value of 140 mg/L, which had higher specificity. It is important to note, however, that the high value demonstrating severe inflammatory reaction is not very relevant in mild or moderate acute pancreatitis; it is more so relevant in due-course necrosis. Readers must be mindful that both thresholds hold key significance in the progression toward an unfavorable course of pancreatitis. PCT’s threshold was 0.5 ng/mL for all the studies. LDH conferred a very high sensitivity finding and may be useful in predicting the progression of acute pancreatitis to pancreatic necrosis. Early and aggressive management of AP has been shown to reduce morbidity and mortality, for which diagnosis and assessment of the severity of the disease are imperative. This paper sought to pinpoint ideal markers in early assessment and prediction of the worsening of the disease. We found that inflammatory biomarkers may be scalable as diagnostic accuracy tests in the clinical progression of acute pancreatitis. There ought to be higher-powered studies with pre-defined thresholds for inflammatory biomarkers to reduce the burden of disease across populations.

## Figures and Tables

**Figure 1 healthcare-11-00027-f001:**
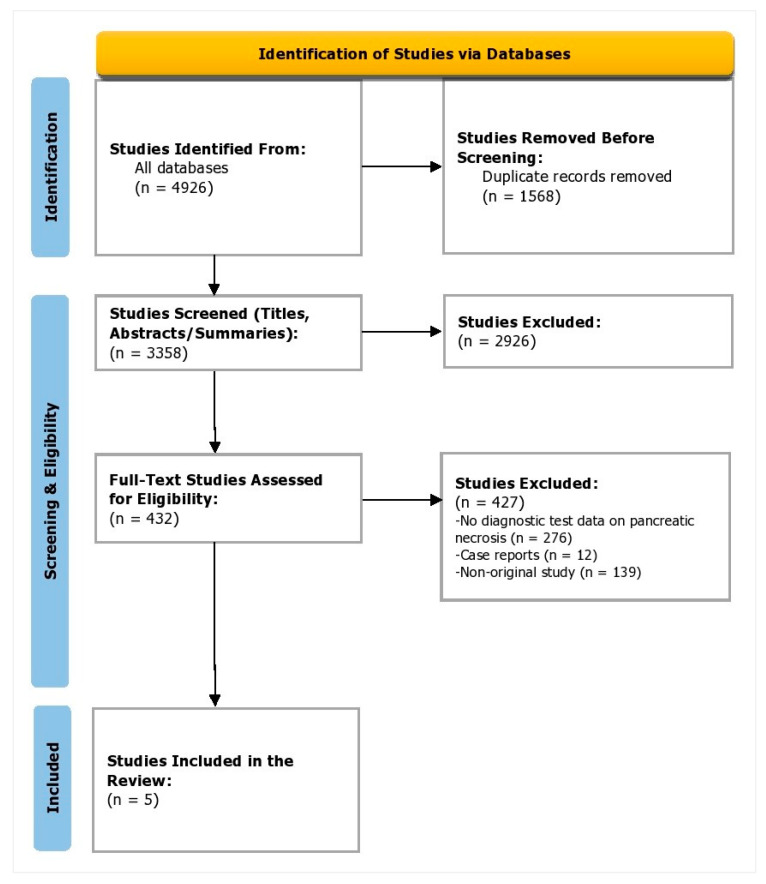
PRISMA flow diagram depicting the study selection process.

**Figure 2 healthcare-11-00027-f002:**
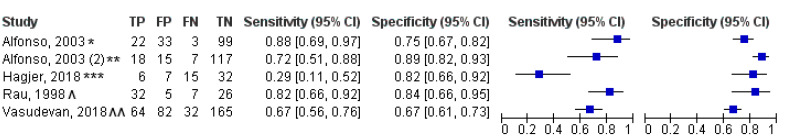
Forest plot of C-reactive protein delineating the sensitivity and specificity of the test. Alfonso, 2003 * (>200 mg/L); Alfonso, 2003 (2) ** (>279 mg/L); Hagjer, 2018 *** (150 mg/L); Rau, 1998 ^ (>140 mg/L); ^^ Vasudevan, 2018 (>98 ng/mL).

**Figure 3 healthcare-11-00027-f003:**
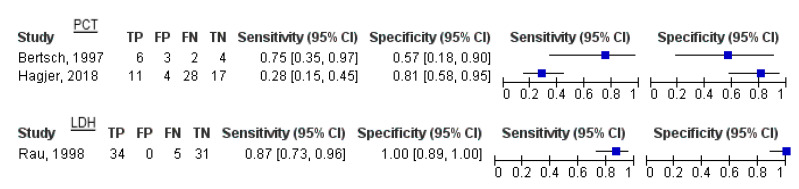
Forest plots of PCT and LDH showcasing the sensitivity and specificity of the tests. The cutoff for PCT was >0.5 ng/mL for both studies. The threshold for LDH was >290 U/L.

**Figure 4 healthcare-11-00027-f004:**
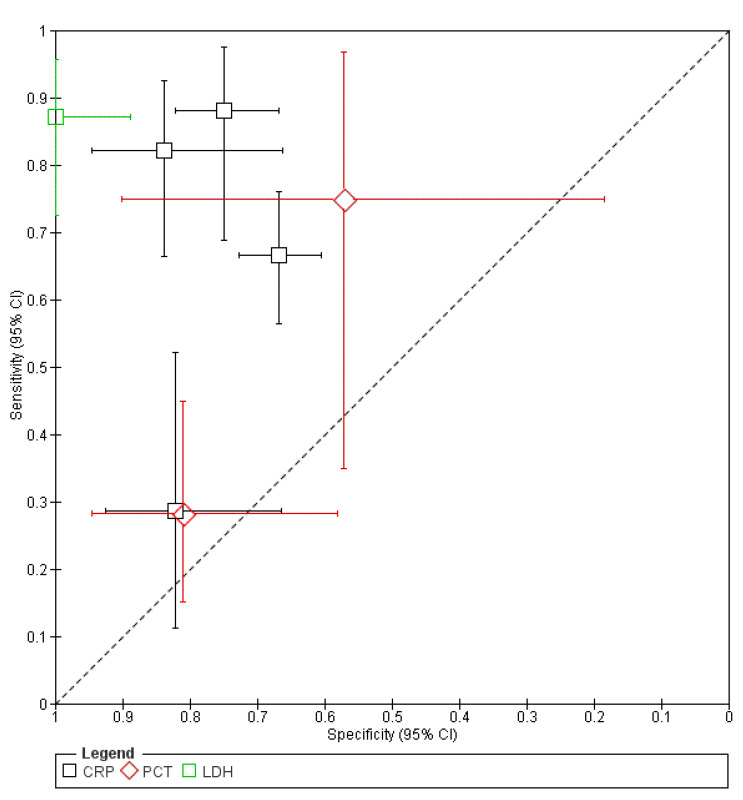
Summary receiver operating characteristic (SROC) plot.

**Figure 5 healthcare-11-00027-f005:**
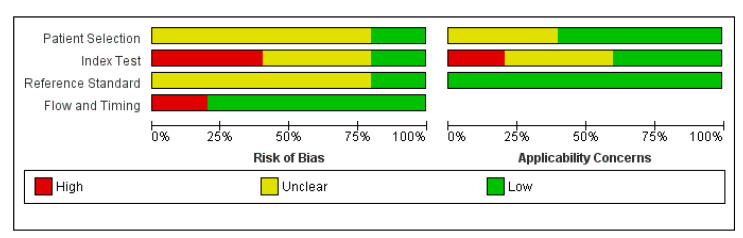
Risk of bias and applicability concerns graph.

**Figure 6 healthcare-11-00027-f006:**
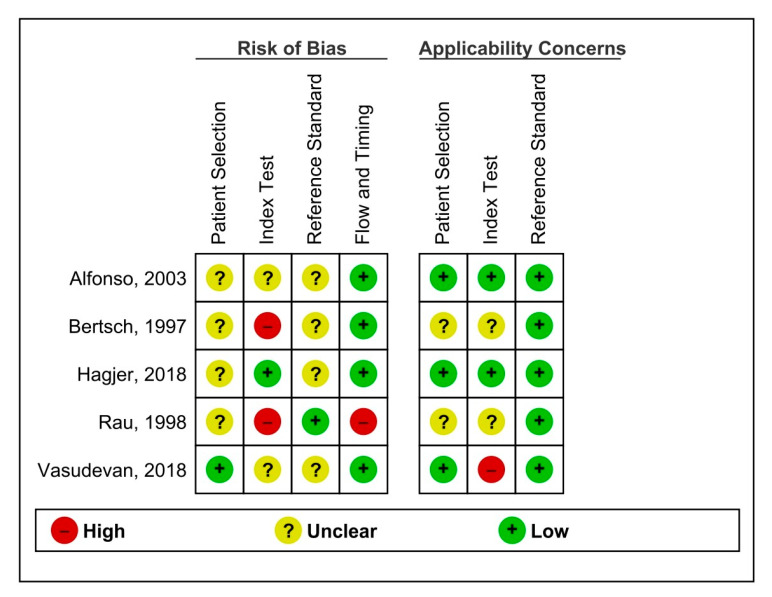
Risk of bias and applicability concerns.

**Table 1 healthcare-11-00027-t001:** Search strategy.

(((((“acute disease”[MeSH Terms] OR (“acute”[All Fields] AND “disease”[All Fields]) OR “acute disease”[All Fields] OR (“biomarker s”[All Fields] OR “biomarkers”[MeSH Terms] OR “biomarkers”[All Fields] OR “biomarker”[All Fields])) AND (“c reactive protein”[MeSH Terms] OR (“c reactive”[All Fields] AND “protein”[All Fields]) OR “c reactive protein”[All Fields] OR “c reactive protein”[All Fields])) OR (“calcitonin”[MeSH Terms] OR “calcitonin”[All Fields] OR “calcitonins”[All Fields] OR “calcitonine”[All Fields]) OR (“l lactate dehydrogenase”[MeSH Terms] OR (“l lactate”[All Fields] AND “dehydrogenase”[All Fields]) OR “l lactate dehydrogenase”[All Fields] OR (“lactate”[All Fields] AND “dehydrogenase”[All Fields]) OR “lactate dehydrogenase”[All Fields]) OR (“pancreas”[MeSH Terms] OR “pancreas”[All Fields] OR “pancreatic”[All Fields] OR “pancreatitides”[All Fields] OR “pancreatitis”[MeSH Terms] OR “pancreatitis”[All Fields])) AND ((“acute”[All Fields] OR “acutely”[All Fields] OR “acutes”[All Fields]) AND (“necrosis”[MeSH Terms] OR “necrosis”[All Fields] OR “necrotic”[All Fields] OR “necrotising”[All Fields] OR “necrotization”[All Fields] OR “necrotize”[All Fields] OR “necrotized”[All Fields] OR “necrotizing”[All Fields]))) OR (“necrose”[All Fields] OR “necrosed”[All Fields] OR “necrosi”[All Fields] OR “necrosing”[All Fields] OR “necrosis”[MeSH Terms] OR “necrosis”[All Fields] OR “necroses”[All Fields])) AND (clinicaltrial[Filter] OR clinicaltrialphasei[Filter] OR clinicaltrialphaseii[Filter] OR clinicaltrialphaseiii[Filter] OR clinicaltrialphaseiv[Filter] OR controlledclinicaltrial[Filter] OR randomizedcontrolledtrial[Filter])
Translations
acute disease: “acute disease”[MeSH Terms] OR (“acute”[All Fields] AND “disease”[All Fields]) OR “acute disease”[All Fields]
biomarkers: “biomarker’s”[All Fields] OR “biomarkers”[MeSH Terms] OR “biomarkers”[All Fields] OR “biomarker”[All Fields]
C-reactive protein: “c-reactive protein”[MeSH Terms] OR (“c-reactive”[All Fields] AND “protein”[All Fields]) OR “c-reactive protein”[All Fields] OR “c reactive protein”[All Fields]
calcitonin: “calcitonin”[MeSH Terms] OR “calcitonin”[All Fields] OR “calcitonins”[All Fields] OR “calcitonine”[All Fields]
lactate dehydrogenase: “l-lactate dehydrogenase”[MeSH Terms] OR (“l-lactate”[All Fields] AND “dehydrogenase”[All Fields]) OR “l-lactate dehydrogenase”[All Fields] OR (“lactate”[All Fields] AND “dehydrogenase”[All Fields]) OR “lactate dehydrogenase”[All Fields]
pancreatitis: “pancreas”[MeSH Terms] OR “pancreas”[All Fields] OR “pancreatic”[All Fields] OR “pancreatitides”[All Fields] OR “pancreatitis”[MeSH Terms] OR “pancreatitis”[All Fields]
acute: “acute”[All Fields] OR “acutely”[All Fields] OR “acutes”[All Fields]
necrotizing: “necrosis”[MeSH Terms] OR “necrosis”[All Fields] OR “necrotic”[All Fields] OR “necrotising”[All Fields] OR “necrotization”[All Fields] OR “necrotize”[All Fields] OR “necrotized”[All Fields] OR “necrotizing”[All Fields]
necrosis: “necrose”[All Fields] OR “necrosed”[All Fields] OR “necrosi”[All Fields] OR “necrosing”[All Fields] OR “necrosis”[MeSH Terms] OR “necrosis”[All Fields] OR “necroses”[All Fields]

**Table 2 healthcare-11-00027-t002:** Characteristics of included studies.

Author, Year	Study Type	Country	Sample Size	Males	Presentation	Inflammatory Biomarkers	Cut-off/Positive Diagnosis	Laboratory Method	Reference Standard Used
Bertsch, 1997 [45]	Cohort study	Germany	15	10 (66.6%)	Acute pancreatitis in the secondary care setting	PCT (ng/mL): Sensitivity = 75% (95% CI = 35–97%); Specificity = 57% (95% CI = 18–90%)	PCT > 0.5 ng/mL	PCT (day 1) tested with a luminometric immunoassay (Fa. Brahms, Berlin)	CT Scan
Rau, 1998 [46,47,48]	Prospective cohort study	Germany	70	39 (55.7%)	Presentation with acute pancreatitis in the secondary care setting in less than 4 days of symptom onset	CRP (mg/L): Sensitivity = 82% (95% CI = 66–92%); Specificity = 84% (95% CI = 66–95%); LDH (U/L): Sensitivity = 87% (95% CI = 73–96%); Specificity = 100% (95% CI = 89–100%)	CRP > 140 mg/L; LDH > 290 U/L	CRP (day 3) was tested with laser nephelometry; LDH (day 5) was tested with the enzyme kinetic method	Intraoperative findings * and/or CT scan (CT 9800 or CT Twin Flash)
Alfonso, 2003 [49]	Retrospective cohort study	Spain	157	94 (59.9%)	Acute pancreatitis in the secondary setting	CRP (mg/L) (1) Sensitivity = 88% (95% CI = 69–97%); Specificity = 75% (95% CI = 67–82%); (2) Sensitivity = 72% (95% CI = 51–88%); Specificity = 89% (95% CI = 82–93%)	CRP: (1) > 200 mg/L (2) > 279 mg/L;	CRP (exact day not stated) with nephelometry (Dade Behring Marburg GmbH, Marburg, Germany)	CT Scan
Hagjer, 2018 [50]	Prospective cohort study	India	60	41 (68.3%)	Acute pancreatitis in the tertiary care setting (referral hospital)	CRP (mg/L): Sensitivity = 29% (95% CI = 11–52%); Specificity = 82% (95% CI = 66–92%); PCT (ng/mL): Sensitivity = 28% (95% CI = 15–45%); Specificity = 81% (95% CI = 58–95%)	CRP > 150 mg/L; PCT > 0.5 ng/ml	PCT-Q test (B·R·A·H·M·S PCT-Q) with Thermo Fisher Scientific within 1 day of presentation	CT Scan
Vasudevan, 2018 [51]	Prospective cohort study	India	343	202 (59%)	Acute pancreatitis in the tertiary care setting	CRP (mg/L): Sensitivity = 67% (95% CI = 56–76%); Specificity = 67% (95% CI = 61–73%)	CRP > 98 mg/L	CRP (on the day of admission)	CT Scan

* The intraoperative findings were determined with laparotomy, which was performed 18 days after the index test was performed; the timing between the CT scan and the index test was 2–6 days.

**Table 3 healthcare-11-00027-t003:** Tabulated summary of meta-analytical findings presenting χ2, *p*-value, PPV, NPV, positive likelihood ratio, and negative likelihood ratio.

Test and Cutoff	χ2	*p*-Value	Positive Predictive Values (PPV)	Negative Predictive Values (NPV)	Positive Likelihood Ratio	Negative Likelihood Ratio
Bertsch, 1997						
PCT > 0.5 ng/mL	1.487	0.223	0.663 (95% CI = 0.413–0.837)	0.669 (95% CI = 0.295–0.93)	1.744 (95% CI = 0.624–4.56)	0.439 (95% CI = 0.067–2.119)
Rau, 1998						
CRP > 140 mg/L	47.425	<0.001 *	0.555 (95% CI = 0.412–0.662)	0.943 (95% CI = 0.906–0.972)	6.545 (95% CI = 3.673–10.29)	0.315 (95% CI = 0.154–0.545)
LDH > 290 U/L	51.596	<0.001 *	1 (95% CI = 0.903–1)	0.86 (95% CI = 0.768–0.86)	Not Estimable	0.13 (95% CI = 0.13–0.24)
Alfonso, 2003						
CRP > 200 mg/L	36.54	<0.001 *	0.401 (95% CI = 0.317–0.441)	0.97 (95% CI = 0.925–0.992)	3.52 (95% CI = 2.44–4.149)	0.16 (95% CI = 0.042–0.426)
CRP > 279 mg/L	47.425	<0.001 *	0.555 (95% CI = 0.412–0.662)	0.943 (95% CI = 0.906–0.972)	6.545 (95% CI = 3.673–10.29)	0.315 (95% CI = 0.154–0.545)
Hagjer, 2018						
CRP > 150 mg/L	0.906	0.341	0.462 (95% CI = 0.219–0.716)	0.681 (95% CI = 0.614–0.751)	1.598 (95% CI = 0.52–4.683)	0.87 (95% CI = 0.615–1.169)
PCT > 0.05 ng/mL	0.631	0.427	0.442 (95% CI = 0.206–0.695)	0.676 (95% CI = 0.609–0.748)	1.474 (95% CI = 0.482–4.24)	0.889 (95% CI = 0.625–1.192)
Vasudevan, 2018						
CRP > 98 ng/mL	31.647	<0.001 *	0.439 (95% CI = 0.38–0.491)	0.838 (95% CI = 0.795–0.877)	2.009 (95% CI = 1.579–2.483)	0.499 (95% CI = 0.362–0.665)

* Statistically significant findings with *p* < 0.05.

## Data Availability

All data obtained for the purpose of this study are available online.

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
