# Peer review of "The Evaluation of Inflammatory Biomarkers in Predicting Progression of Acute Pancreatitis to Pancreatic Necrosis: A Diagnostic Test Accuracy Review"

_healthcare, 2022, doi:10.3390/healthcare11010027_

Round 1
Reviewer 1 Report (New Reviewer)
Keep up the good work. Please see attached word file.

Author Response
Reviewer 1 Comments and Author Responses:
To the authors:
We command the authors for a review that is well thought off and writing which is authentic. Keep up the good work. We found the following points that need careful corrections.
Page 1
Line 1, in the title the capital (I) in In needs to be small (in).
Author response: Thank you for your comment. It has been updated.
Abstract the abbreviation CINHAL needs to be define as CINAHL Cumulative Index of Nursing and Allied Health (CINAHL) then reused again in page 2.
Author response: Thank you for your comment. Your comment is valuable to us and we have changed them in both places.
Page 2
You wrote (This diagnostic test accuracy review was conducted in accordance with The Preferred Reporting Items for Systematic Reviews and Meta-Analysis
(PRISMA) 2020 Checklist [39].) it should read (This diagnostic test accuracy review was conducted in accordance with the Preferred Reporting Items for Systematic Reviews and Meta-Analysis (PRISMA) 2020 Checklist [39].)
Author response: Thank you for your comment. It has been updated.
Line 94-96 you wrote (Study searches were conducted across the following databases: PubMed (MEDLINE), CINAHL Plus (EBSCO host), EMBASE (Elsevier), Cochrane Central (Wiley Online Library), and Web of Science (Clarivate Analytics) until March 31st, 2022.). Then you wrote in page 4, line 115 (The cutoff is typical in diagnostic test accuracy reviews, as recommended by the Cochrane Handbook [40].) Which you access on May, 30th, 2022! References 40, 41, and 42 were all accessed on May 2022, so that, why the searches were NOT conducted across the following databases: PubMed (MEDLINE), CINAHL Plus (EBSCO host), EMBASE (Elsevier), Cochrane Central (Wiley Online Library), and Web of Science (Clarivate Analytics) until May 30th, 2022?
Author response: Thank you for your comment. We had conducted the first and final search until March 31st, 2022. At the time of paper writing, we conducted an additional search of all databases on May 30th, 2022 and did not find any relevant studies. I am making the change in the manuscript to ensure we write correct dates.
Page 3 pages 103 & 105
You wrote: (The search strategy for PubMed is appended in Supplementary Table
- No time restrictions were applied to ensure all relevant studies were located.) There was not a Supplementary Table included in the review!
Author response: Thank you for your comment. Please review “Table 1.” It is highlighted in green and moved to the main manuscript.
Lines 115 to 129 your references 40, 41, and 42 are all accessed on May 30th, 2022 see the point raised above on page 2.
Author response: Thank you for your comment. We had conducted the first and final search until March 31st, 2022. At the time of paper writing, we conducted an additional search of all databases on May 30th, 2022 and did not find any relevant studies. I am making the change in the manuscript to ensure we write correct dates.
Page 4
See the point in page 2 mentioned above.
Author response: Thank you for your comment. It has been fixed.
Pages 5-7 including Table 1
Lines 177, 178 you wrote: (In total, fives studies were included in this analysis: Bertsch (1997) [45], Rau (1998) [46 48], Alfonso (2003) [49], Hagjer (2018) [50],
Vasudevan (2018) [51]. Five studies pooling 645 participants). We did not understand anything from these writings! What are these fives studies mentioned? Then, you stated 7 references! [45-]
names for these 7 references but the paragraph is NOT informative at all.
Table 1 is clear and understandable, but the writing is not matching the numbers in Table 1.
Author Response: I have made additions and changes to the writing. This has been clarified to the following: The section 3.2. Overview of Included Studies, has been updated with an overview of all the included studies. I urge you to see the new text. I hope that provides the relevant information.
Page 8, under Table 2
Why not using the proper symbol for Chi square?
Author response: The text writing has been updated with the symbol.
Page 9-13
Figure 2 to Figure 6 are all good, but very specialized members of the journal audience will understand these figures.
Author response: We have tried our best to write them down with findings understandable the regular scientific reader. However, as with such studies, some understanding of the methods used is required.
Discussion and conclusion
Line 288, you wrote (However, the WCES guidelines state concerns over post- contrast acute kidney injury (AKI) [54].) It should read (However, the WCES guidelines state concerns over post-contrast acute kidney injury [54].)
Author response: Thank you for your comment. It has been updated.
You defined and abbreviated AKI. You usually do that if you are going to reuse the abbreviation somewhere else in the manuscript. You do not need to abbreviate acute kidney injury.
The rest of this section is all good.
Author response: Thank you for your comment. It has been fixed as mentioned above.
References are good. However, why there are more than one style in citing the references? As in ref. 1 and ref. 3 compared to ref. 4. & ref. 6 in the first group references you gave the page range while you just gave the first page number only? Be consistent and abide by the guide for author’s for citing the references.
below:
- Swaroop, V.S.; Chari, S.T.; Clain, J.E. Severe Acute Pancreatitis. Jama 2004, 2865 2868. [Page range was given]
Banks, P.A. Freeman ML, Practice Parameters Committee of the American College of Gastroenterology. Pract. Guidel. acute pancreatitis. Am J Gastroenterol 2006, 101,
2379 2400. [Page range was given] Shah, A.P.; Mourad, M.M.; Bramhall, S.R. Acute Pancreatitis: Current Perspectives on Diagnosis and Management. J. Inflamm. Res. 2018, 11, 77. [Page range was not given]
- Johnson, C.D.; Besselink, M.G.; Carter, R. Acute Pancreatitis. Bmj 2014, 349. [Page range was not given]
Also, in references 29 and 30 and the very last ref. #57you are missing the page ranges, for example, ref # 29 the page range is: 1973 1985. Please go over all the references and include the correct page ranges.
Author response: Thank you for your comment. All discrepancies in the references have been fixed. Do note however that reference number 6 does not have page ranges as is it electronic only.
It is stated as follows, and there is no alternative: Johnson, C.D.; Besselink, M.G.; Carter, R. Acute Pancreatitis. Bmj 2014, 349.
Howewer, ALL OTHER CHANGES HAVE BEEN MADE AND ARE HIGHLIGHTED IN GREEN FOR YOUR REVIEW.
All Tables are good.
All figures are good.
Best of Regards,
Author Comments for the Reviewer:
To the esteemed reviewer, thank you for allowing us to make exponential improvements with our study. I thank you for the time and attention you have given to our paper.
Regards,
Dr. Zouina Sarfraz
Reviewer 2 Report (New Reviewer)
The present article, entitled, "The Evaluation of Inflammatory Biomarkers In Predicting Progression of Acute Pancreatitis to Pancreatic Necrosis: A Diagnostic Test Accuracy Review" by Riaz et al., is a well written and comprehensively described work, that focuses on assessment of the diagnostic accuracy of inflammatory biomarkers: CRP, PCT, and LDH. Here is my comment:
The quality of Figure 1, 3 and 6 is poor. Authors are advised to replace the mentioned figures with a better one, respectively.
Author Response
Reviewer 2 Comments and Author Responses:
Comments:
The present article, entitled, "The Evaluation of Inflammatory Biomarkers In Predicting Progression of Acute Pancreatitis to Pancreatic Necrosis: A Diagnostic Test Accuracy Review" by Riaz et al., is a well written and comprehensively described work, that focuses on assessment of the diagnostic accuracy of inflammatory biomarkers: CRP, PCT, and LDH. Here is my comment:
The quality of Figure 1, 3 and 6 is poor. Authors are advised to replace the mentioned figures with a better one, respectively.
Author Response: I have updated them to 1200 dpi, which is the highest available quality from the software. These updated figures have been reattached to the document and I will email them to the editorial team when typesetting the PDF. That ensures best quality figures. Thank you for raising this concern!
Author Comments for the Reviewer:
To the esteemed reviewer, thank you for allowing us to make exponential improvements with our study. I thank you for the time and attention you have given to our paper.
Regards,
Dr. Zouina Sarfraz
Reviewer 3 Report (New Reviewer)
I think that a huge work was done starting with 3358 published papers of which 432 were eligible with surprisingly, only 5 studies to be analyzed (two were before 2000 year).
Line 194 the value of 98 mg/mL should be corrected to mg/L.
Table 1. Inflammatory markers are presented with True positive, True negative etc values which is not useful since such a presentation is difficult to follow and not relevant. I suggest to replace with Sensitivity, Specificity, PPV, NPV, Youden index.
Rau et al. and Hagjer et al. studies yielded similar PCR thresholds upon 70 and 60 patients, respectively but the sensitivity was very low in the Rau study raising questions about the reliability of measuring or timing.
Figure 4 may be better presented in the well known format as Receiver Operating Curves.
Lines 335-337 The cutoff with optimal CRP value was 200 mg/L which
had higher sensitivity...... High value demonstrating severe inflammatory reaction not very relevant in a mild or moderate acute pancreatitis, probably in due course necrosis. It seems to be rather late to confirm an unfavorable course of pancreatitis if using this threshold.
Author Response
Reviewer 3 Comments and Author Responses:
Comment 1: I think that a huge work was done starting with 3358 published papers of which 432 were eligible with surprisingly, only 5 studies to be analyzed (two were before 2000 year).
Author response to comment 1: Dear reviewer, we regularly conduct systematic reviews and as a team, a pair of two researchers divide the studies and check through all studies for inclusion. This is a common occurrence and justification for the study selection process has been listed. For any other queries, feel free to reach out please.
Comment 2: Line 194 the value of 98 mg/mL should be corrected to mg/L.
Author response to comment 2: Thank you for your due attention. It has been corrected.
Comment 3: Table 1. Inflammatory markers are presented with True positive, True negative etc values which is not useful since such a presentation is difficult to follow and not relevant. I suggest to replace with Sensitivity, Specificity, PPV, NPV, Youden index.
Author response to comment 3: Thank you for your comment. The TP, TN, etc. values have been removed and have been replaced with sensitivity and specificity. Please review the changes in green. The table 3 has enlisted values of PPV, NPV, etc. and has been separated from Table 2.
Comment 4: Rau et al. and Hagjer et al. studies yielded similar PCR thresholds upon 70 and 60 patients, respectively but the sensitivity was very low in the Rau study raising questions about the reliability of measuring or timing.
Author response to comment 4: A note has been made in the limitations section:
“With Rau et al. and Hagjer et al.’s studies yielding similar PCT thresholds upon 70 and 60 patients, respectively, the sensitivity was significantly lower in the study by Rau et al.; there are limitations about the reliability of measuring or timing.”
Comment 5: Figure 4 may be better presented in the well known format as Receiver Operating Curves.
Author response to Comment 5: The figure is already presented as a summary ROC plot. Please review the figure for any discrepancies.
Comment 6: Lines 335-337 The cutoff with optimal CRP value was 200 mg/L which
had higher sensitivity...... High value demonstrating severe inflammatory reaction not very relevant in a mild or moderate acute pancreatitis, probably in due course necrosis. It seems to be rather late to confirm an unfavorable course of pancreatitis if using this threshold.
Author response to Comment 6: The conclusion has been added with the following statement to ensure we account for your very important concern:
“It is important to note, however, that the high value demonstrating severe inflamma-tory reaction is not very relevant in a mild or moderate acute pancreatitis; it is more so relevant in due course necrosis. Readers must be mindful that both thresholds hold key significance in the progression towards an unfavorable course of pancreatitis.”
Author Comments for the Reviewer:
To the esteemed reviewer, thank you for allowing us to make exponential improvements with our study. I thank you for the time and attention you have given to our paper.
Regards,
Dr. Zouina Sarfraz
Round 2
Reviewer 1 Report (New Reviewer)
Good Work, keep it up.
Reviewer 3 Report (New Reviewer)
I think the paper is better now.
This manuscript is a resubmission of an earlier submission. The following is a list of the peer review reports and author responses from that submission.
Round 1
Reviewer 1 Report
Dear editors. I congratulate you on your manuscript. The work has been carried out in an excellent manner, however, it does not bring great news from the scientific point of view.
Author Response
Note to Reviewer:
Before you review the revised paper, please make note that the paper has been proofread in this review phase by two English native writers/speakers. Furthermore, 17 new references have been added with updates made throughout the paper. The figures have been made clearer and so have the tables for readability. I anticipate a positive response!
Comments, Reviewer 1:
Dear editors. I congratulate you on your manuscript. The work has been carried out in an excellent manner, however, it does not bring great news from the scientific point of view.
Author Responses to Reviewer 1: Thank you for your comments. We provide hope for the patient and healthcare worker community with concluding remarks and strengths of the study revised. I anticipate your positive response on our revised paper.
Reviewer 2 Report
1. Abstract is helping readers to know the paper’s research and findings with a concise and conclusive description. Please remove the “Backgrounds” “Methods”, “Results” and “Conclusions” words in abstract. Make the abstract be more conclusive.
2. Line 160, add one “+” in the middle of “True Negatives” and “False Negatives”.
3. Please adjust the format of Table 1. For example, make column 3 be wider for showing the word at one line.
4. Please improve the resolution of Figures 2,3,5, especially the bold words in the figures.
Author Response
Note to Reviewer:
Before you review the revised paper, please make note that the paper has been proofread in this review phase by two English native writers/speakers. Furthermore, 17 new references have been added with updates made throughout the paper. The figures have been made clearer and so have the tables for readability. I anticipate a positive response!
Comments, Reviewer 2:
Comment 1. Abstract is helping readers to know the paper’s research and findings with a concise and conclusive description. Please remove the “Backgrounds” “Methods”, “Results” and “Conclusions” words in abstract. Make the abstract be more conclusive.
Author Response: To the Reviewer, I thank you very much for this helpful comment. Your insights are greatly appreciated in improving our paper for publication.
I have revised the abstract entirely; Please review the revised abstract:
“Acute pancreatitis is the acute inflammation of the pancreas; 30% of cases may progress to pancreatic necrosis. The aim of this study was to assess the diagnostic accuracy of inflammatory bi-omarkers (C-reactive protein (CRP), procalcitonin (PCT), and lactate dehydrogenase (LDH)) in detecting pancreatic necrosis in adults with confirmed acute pancreatitis within 14 days of symp-tom onset and without organ failure. A systematic search was conducted across CINAHL, Cochrane, Embase, PubMed, and Web of Science until March 31, 2022, with the following key-words: acute disease, biomarkers, C‐reactive protein, calcitonin, differential, diagnosis, lactate dehydrogenase, pancreatitis, acute necrotizing, necrosis, sensitivity, specificity. Statistical analysis was conducted in RevMan 5.4.1 (Cochrane). Five studies pooling 645 participants were included of which 59.8% were males, with a mean age of 49 years. CRP was best cut off at 279 mg/L (χ2= 47.43, P<0.001), followed by 200 mg/L (χ2= 36.54, P<0.001). LDH was cutoff at 290 units/L (χ2= 51.6, P<0.001), whereas PCT did not display the most reliable results at 0.05 ng/mL. Inflammatory biomarkers are scalable diagnostic tools that may confer clinical value by decreasing mortality of acute pancreatitis sequelae.”
Comment 2. Line 160, add one “+” in the middle of “True Negatives” and “False Negatives”.
Author Response: Thank you for noting the error. It has been fixed.
Comment 3. Please adjust the format of Table 1. For example, make column 3 be wider for showing the word at one line.
Author Response: Thank you for your much needed suggestion. I have updated the formatting to landscape, which will help readers in overcoming the stuffiness in the paper. I hope that resolves the issue! :)
Comment 4. Please improve the resolution of Figures 2,3,5, especially the bold words in the figures.
Author Response: The figures have been generated again at the highest quality resolution. These are usually improved when the PDF is generated for publication but I hope you see the improvement with the newer quality figures now. Thank you for your due diligence.